# Molecular Cavity Topological Representation for Pattern Analysis: A NLP Analogy-Based Word2Vec Method

**DOI:** 10.3390/ijms20236019

**Published:** 2019-11-29

**Authors:** Dongliang Guo, Qiaoqiao Wang, Meng Liang, Wei Liu, Junlan Nie

**Affiliations:** 1School of Information Science and Engineering, Yanshan University, Qinhuangdao 066004, China; dongliangguo@ysu.edu.cn (D.G.); wangqiaoqiao92@163.com (Q.W.); liangmengxa@163.com (M.L.); 2Key Laboratory of Software Engineering in Hebei Province, Qinhuangdao 066004, China; 3Advanced Analytics Institute, Faculty of Engineering and IT, University of Technology Sydney, Ultimo NSW 2007, Australia; Wei.Liu@uts.edu.au; 4Key Laboratory of Computer Virtual Technology and System Integration in Hebei Province, Qinhuangdao 066004, China

**Keywords:** molecular cavity, topological representation, Word2Vec model, analogy-based methods

## Abstract

Cavity analysis in molecular dynamics is important for understanding molecular function. However, analyzing the dynamic pattern of molecular cavities remains a difficult task. In this paper, we propose a novel method to topologically represent molecular cavities by vectorization. First, a characterization of cavities is established through Word2Vec model, based on an analogy between the cavities and natural language processing (NLP) terms. Then, we use some techniques such as dimension reduction and clustering to conduct an exploratory analysis of the vectorized molecular cavity. On a real data set, we demonstrate that our approach is applicable to maintain the topological characteristics of the cavity and can find the change patterns from a large number of cavities.

## 1. Introduction

Proteins are highly complex systems. The structures of proteins, such as surfaces and internal cavities, directly determine the behavior and vital activity of the species. The function of a macromolecule often requires that a small molecule or ion is transported through these structures. Transport pathways play an essential role in the functioning of a large number of proteins. The best known include (i) uncovering the principles of life [1], (ii) the development of new drugs [2], and (iii) applications in industry [3]. Identification of cavities in the complex protein structures is a difficult challenge, and many software tools have been developed recently for this purpose, e.g., CAVER [4], MOLE [5], MolAxis [6], ChExVis [7], or BetaCavityWeb [8]. All these tools are based on computational geometry methods that employ the Voronoi diagrams. A set of cavities computed by CAVER and visualized using software PyMOL [9] can be seen in Figure 1.

Properties of transport pathways changed significantly over time in molecular dynamics simulation, the analysis of an ensemble of structures, in contrast to a single static structure, is needed for the understanding [4,10]. With the current computational capacity, it becomes affordable to obtain molecular dynamics (MD) trajectories up to the microsecond time scales. This trend requires new approaches to explore the large data sets, as it becomes impracticable to observe such simulation in a frame-by-frame manner.

Characteristics of individual cavities, such as geometry, physico-chemical properties, and dynamics, are instrumental for identifying and analysing cavities [4]. The existing representation of cavities is focusing on geometry representation. Geometry representation contains only a few topological properties and is not sufficient to directly support the analysis of topological properties of molecular cavities. To fill this gap, we propose a vectorized representation method based on cavity atomic sequences, which allow combining the topological attribute to analyze very large sets of conformations containing hundreds of thousands of cavities. By analyzing the dynamic pattern of molecular cavities, it can be inferred which amino acids or atoms play a key role in different patterns of variation causing differences in cavities shape and bottleneck. This may be helpful in understanding the molecular function or developing new drugs.

Our approach builds on several aspects of previous work on molecule cavity and distributed representation. We have divided the related work into two sections accordingly.

### 1.1. Molecular Cavity Computation and Analysis

Many methods have been developed for cavity extraction and clustering. Almost all algorithms for the extraction of cavities simplify the molecule use the hard sphere model. Krone et al. [11] classified such methods into four main categories by the computational methods used for the cavity detection. The four main categories are formed by methods based on grids, Voronoi diagrams, molecular surfaces, and probes. The first algorithm to detect cavities in molecules was based on grids. This algorithm is not suitable for larger structures due to hardware limitations. Voronoi diagram has proved [12] to be suitable for detection of paths in molecules and was able to process large molecules as well.

The Voronoi diagram method can be used for processing of molecular dynamics trajectories. The protein is often simplified by the atom positions or the hard sphere model. Moreover, the edges of Voronoi diagrams automatically provide geometrically optimal molecular paths based on the restriction. Because of the lining atom sequence is easier to achieve, a Voronoi-based method is adopted to extract cavity in our work.

To estimate the distance of the two pathways, a method was proposed that evaluated the distance between each ball from the first pathway and its respective closest ball from the second pathway. To accelerate the computation of all pairwise pathway similarities, Eva Chovancova et al. [4] designed a more efficient algorithm. The basic idea is to represent each pathway by a sequence of *N* points and then compute only distances between pairs of points with the corresponding order, thus performing only *N* distance computations for each pair of pathways.

For clustering pathways, two main methods are currently used. Clustering of pathways in CAVER 3.0 [4] was based on the precalculated distance matrix. Average-link hierarchical clustering [13] was used to construct a tree hierarchy of pathway axes based on their pairwise distances. MOLE 1.2 [14] offered the most similar functionality to work presented in this paper. It employs clustering for finding the correspondence between cavities from different conformations. The similarity of cavities was computed by comparing the sets of atoms lining the cavities. Experiments revealed that the clustering depends on the ordering of the cavities. However, the clustering of cavities performed by MOLE 1.2 [14] was not able to clearly separate cavities into clusters corresponding to known transport pathways. To improve the cluster result, we propose a vectorized representation based on molecular cavity atomic sequences, which adds topological characteristics.

### 1.2. Distributed Representation

Distributed representation has been proven to be one of the most successful methods of machine learning [15,16]. The premise of this approach is encoding and storing information about an item within a system through establishing its interactions with other members. Distributed representation was originally inspired by the structure of human memory, where the items are stored in a ”content-addressable” fashion [17].

Distributed vector representation has been recently established in natural language processing (NLP) as an efficient way to represent semantic/syntactic units with many applications. In this model, known as word2vec, each word is embedded in a vector in a n-dimensional space. Similar words have close vectors, where similarity is defined in terms of both syntax and semantic. The basic idea behind training such vectors is that the meaning of a word is characterized by its context, i.e., neighboring words. Thus, words and their contexts are considered to be positive training samples. Such vectors can be trained using large amounts of textual data in a variety of ways, e.g., neural network architectures like the Skip-gram model [16]. Interesting patterns were observed by training word vectors using Skip-gram in natural language. Similar words can have multiple degrees of similarity. Using a word offset technique where simple algebraic operations are performed on the word vectors, it was shown for example that vector(“King”) − vector(“Man”) + vector(“Woman”) results in a vector that is closest to the vector representation of the word Queen [18].

The Word2Vec algorithm can be applied in various NLP tasks including sentiment analysis, entity linking, machine translation and so on. It can also applies in a variety of other fields [19]. Berger et al. [20] designed a visualization scheme named Cite2Vec to support a dynamically exploration of document collections. Based on an analogy between transportation elements and NLP terms, the Word2Vec model can also be employed to quantify the implicit traffic interactions among roads [21]. Similar attempts have been made in the biomedical field to represent data in a continuous vector space. ProtVec [17] applies Word2Vec to a protein sequence to obtain distributed representations of a 3-gram amino acid sequence. The protein sequence is initially split into 3-gram each having a biological significance and regarded as a “word”. The next step is to run the Word2Vec algorithm using Skip-gram. Seq2Vec [22] extends ProtVec’s approach by applying not only the sequential elements of ProtVec 3-gram, but also the Doc2Vec representation sequence by directly embedding the sequence itself. Dna2Vec [23] generalizes the 3-gram structure of ProtVec and Seq2Vec to the k-gram structure. Another method involves SNP2Vec [24], which embeds a single SNPs into a contiguous space by using a denoising autoencoder [25] and Diet Networks. Mut2Vec [26] can be utilized to generate distributed representations of mutations and experimentally validate the efficacy of the generated mutation representations. It also can be used in various deep learning applications such as cancer classification and drug sensitivity prediction.

Similarly, in the field of molecule cavity analysis, a cavity sequence can be encoded by its lining atoms. To the best of our knowledge, Word2Vec and Doc2Vec have not been used to represent molecule cavity sequences. We propose a novel pipeline to generate distributed representations of topological features for the characterization of molecule cavity. These embeddings are low-dimensional and can greatly simplify downstream modeling. We encode cavity lining atoms as words and lining atom sequences as sentences. A Word2Vec model is further applied to obtain a distributed representation of cavity lining atom sequences. To illustrate, we specifically tackle visualization and cavities classification problems.

This paper is organized as follows. In the next section, we elaborate on the method used in the experiment in Section 2. In Section 3, we mainly introduce the experimental results. Finally, we provide some discussions and conclusions in Section 4.

## 2. Methods

Our molecule cavity distributed representations generate pipeline is summarized as follows. First, we calculate and extract molecule cavity sequences from PDB ( Protein Data Bank) [27] with a Voronoi-based method. Next, we train the cavity node vectors using the Skip-gram model. Finally, we use cavity node vectors generate cavity sequence vectors. The whole pipeline is described in Figure 2.

### 2.1. Cavity-Space Construction

Our method is to construct a distributed representation of cavity sequences. In the training process of word embedding in NLP, a large corpus of sentences should be fed into the training algorithm to ensure sufficient contexts are observed. Similarly, a large corpus is needed to train distributed representation of cavity sequences. In our work, the data of DhaA80 [28] be used as test data, which includes 8052 cavity trajectories.

For the representation of a cavity, a set of atoms surrounding the cavity should be calculated. These atoms are often denoted as the lining atoms. There are several methods for the computation of the lining atoms. Almost every solution for cavities computation uses its own method for the detection of these atoms. For example, MOLE 2.0 [5] samples the cavity centerline where the distance between two samples is set at 0.1 angstrom. In each sample, the five closest atoms are detected, and these atoms can be marked as the lining ones. In our case, we use the approach of CAVER 3.0 [4], which is based on Voronoi. For convenience, we employ the data produced using the CAVER Analyst 2.0 tool [28]. Here, the atoms are defined as cavity lining atoms if their distance to the cavity centerline is the smallest four atoms. It is possible to use any other algorithm for the detection of cavity lining atoms. Our method can be easily adjusted to any of the existing solutions. It requires as input only the list of spheres forming the cavity body and the list of surrounding atoms and amino acids with additional information, such as their spatial orientation, type, or physico-chemical properties. Cavity spheres are spheres of maximal radii with respect to the surrounding atoms and are positioned on the cavity centerline. Figure 3 presents an example of a resulting set of residues lining the computed cavity (gray) and the lining atoms (red).

The pathway of a static cavity can be defined in a Voronoi diagram. A Voronoi vertex in this pathway has spatial attributes, such as the practical position. It is known from computational geometry that a Voronoi vertex is determined by four non-coplanar points in 3D space. In our case, these points are the centers of four closest atoms. Therefore, the topological attribute and the adjacent atoms are natively contained in the cavity. Using the Voronoi vertex as a node in the graph, we can obtain a path from an active site to the outer surface. The cavity representation is shown in Figure 4. These nodes are sorted, and the four closest atoms are used to identify the various nodes. To determine the uniqueness of a node, we sort the atom IDs from low to high as the node ID. The molecular cavity sequence is described as
(1)$$x=\langle x_{1},x_{2},\cdots x_{n}\rangle ,x_{i}=\langle a_{1},a_{2},a_{3},a_{4}\rangle \left(a_{1}\leq a_{2}\leq a_{3}\leq a_{4}\right)$$
where *x* is cavity sequence, $x_{i}$ is the node, *n* represents the number of nodes, and *a* is the atom ID. i.e., “1.2.3.4”, “2.3.1.4”, and “3.4.2.1” will be one node ID denoted “1.2.3.4”. The pathway points of the cavity are generated by Voronoi vertices. As a node, each pathway point distinguishes itself by its node ID.

Word-embedding models represent words in a vector space where semantically similar words are mapped to nearby points. The hypothesis behind these models states that words that appear in the same contexts share semantic meaning. A textual document is composed of sequential words, whereas a cavity sequence is composed of sequential node’s ID. Regarding each node ID as a word and each cavity sequence as a document, we can obtain the vector representations of node’s ID by training plenty of cavity sequences using the word embedding models. Correspondingly, high similarity between two cavities vectors indicates the two cavities have greater similarity in topological features.

In our experiment, Word2Vec is used to train the distributed representations of similarity. The Word2Vec [16,18] includes two model architectures: one is the continuous bag-of-words model (CBOW), and the other is the continuous skip-gram model (Skip-gram). CBOW uses a word’s context words in a surrounding window to predict the word, while Skip-gram uses a word to predict its surrounding words. Algorithmically, the above two models are similar, however CBOW performs better on syntactic prediction and the Skip-gram model is better than CBOW on semantic. We need to extract semantic information of cavity. Therefore, we draw on the Skip-gram model to train the contextual similarity of molecular cavity node structures in our work.

When generating distributed vectors for words, the Skip-gram model [16] captures the semantic and grammatical information among words using the slide windows. Given a sequence of training words $\omega_{1}$, $\omega_{2}$, $\omega_{3}$, …, $\omega_{T}$, the objective of the Skip-gram model is to maximize the average log probability:(2)$$\frac{1}{T}\sum_{t=1}^{T}\sum_{-c\leq j\leq c,j\neq 0}\log p\left(\omega_{t+j}\left|\omega_{t}\right.\right)$$
where *c* is the size of the training context (which can be a function of the center word $\omega_{t}$). Larger *c* results in more training examples and thus can lead to a higher accuracy, at the expense of the training time. The basic Skip-gram formulation defines $p\left(\omega_{t+j}\left|\omega_{t}\right.\right)$ using the softmax function
(3)$$p\left(\omega_{O}\left|\omega_{I}\right.\right)=\frac{\exp\left({{v^{\prime }}_{\omega_{O}}}^{T}v_{\omega_{I}}\right)}{\sum_{\omega =1}^{W}\exp\left({{v^{\prime }}_{\omega }}^{T}v_{\omega_{I}}\right)}$$
where $V_{\omega }$ and $V_{\omega }^{\prime }$ are, respectively, the “input” and “output” vector representations of ω, and *W* is the number of words in the vocabulary. This formulation is impractical because the cost of computing $\nabla \;\log p\left(\omega_{O}\left|\omega_{I}\right.\right)$ is proportional to *W*, which is often large.

In our work, we construct cavity embeddings based on node embeddings [30,31]. We average the vectors of the nodes in one cavity $x=\langle x_{1},x_{2},\cdots x_{n}\rangle$ to obtain cavity embeddings (called cavity2vec). The main process in this step is to learn the node embedding matrix $W_{\omega }$:(4)$$v_{cavity2vec}\left(x\right)=\frac{1}{n}\sum_{i}^{n}W_{\omega }^{x_{i}}$$
where $W_{\omega }^{x_{i}}$ is the node embedding for node $x_{i}$, which could be learned by the Word2Vec algorithm.

### 2.2. Dimension Reduction and Cavity Classification

It is a reasonable way to conduct a direct simplification strategy in the original high-dimensional space. However, it is a difficult task to simplify high-dimensional data items by means of traditional sampling strategies [32], especially when the representation of a cavity is depicted with more than 100 dimensional vectors obtained by Word2Vec model, which is well over the capability of human visual cognition. As a commonly used dimension reduction method, t-Distributed Stochastic Neighbor Embedding (t-SNE), proposed by Maaten and Hinton [33], can be applied to reduce multiple dimensions of vectorized representations and visualize the cavities with two-dimensional points, as the t-SNE is capable of capturing much of the local structure while also revealing the global structure of data items. The distance between each pair of points largely depicts the correlation between cavities in the high-dimensional semantic space. In other words, the cavities would be similar when two points are located close to each other.

All cavity trajectories from conformations are clustered to allow the statistical analysis of the properties of corresponding cavities, i.e., providing an opportunity to study the dynamics of cavities. Using the clustering method, the count of clusters in cavity trajectories can be analyzed. A collection of trajectories of a certain cavity under different snapshots, forming a cluster of cavity trajectories. Cluster analysis is built on similarity. The similarity between patterns in a cluster is greater than the similarity between patterns that are not in the same cluster. In our work, the similarity measure using cosine similarity.

Two clustering algorithms are mainly used. The K-means algorithm [34] is the most classical clustering method. As Table 1 shows, the biggest advantages of this algorithm are simplicity and speed. The downside is that how many classes of data should be known in advance. Hierarchical clustering [35] is another major clustering method. This method is a prototype-based clustering algorithm, which attempts to divide the data set at different levels to form a tree-like clustering structure. The data set can be divided into a bottom-up aggregation strategy or a top-down split strategy. The advantage of the hierarchical clustering algorithm is that we can use a dendrogram to help us interpret the clustering results using a visual approach. Another advantage of hierarchical clustering is that it does not require the number of clusters to be specified in advance.

In different application scenarios of our work, we use the two clustering algorithms mentioned above to cluster the molecular cavity as needed. Because the count of clusters of aggregate is unknown, hierarchical clustering algorithm can be used to analyse the amount of clusters in all cavity trajectories. When the amount of cavity change patterns can be inferred by the t-distributed stochastic neighbor (t-SNE) method, the K-means method can be used to analyse the pattern of a certain cavity changes over time.

## 3. Results

To explore the semantic properties of cavities, a clustering view is used to show the differences among clusters in the high-dimensional cavity embedding space. In the natural course of events, clustering results are displayed in two dimensional coordinates. We need to reduce the dimensions of the high-dimensional word embeddings into two-dimensional coordinates to show the structure of clusters. Here, the t-SNE method be used in the clustering view.

The attributes of clustering can be observed from the clustering view. From the quality of the clustering, such as the closeness of clustering and the distance between clusters (overlap of clusters), whether the distributed representation in the space is reasonable, so as to explore the cluster relationship of high-dimensional cavity embedding space, is unknown. As Figure 5 shows, 8052 cavity trajectories are embedded in the 2D view and, the closer two dots are, the more similar they are in the attributes. We intuitively see there are hundreds of clusters. The clustering result of CAVER Analyst 2.0 [28] includes 246 clusters. Therefore, it can be concluded that the vectorized representation of the cavity retains the semantic information and the semantically similar cavities are closer together in the t-SNE clustering view.

Two cavities are considered to be similar if at least some of the two cavities have the same lining atoms and similar topological properties. After training the model, we measure the similarity between two cavities by calculating the cosine similarity of their vectors. Given two cavity vectors, $v_{1}$ and $v_{2}$, the cosine similarity, cos(θ), is represented using a dot product and magnitude as follows,
(5)$$\mathrm{Cos}Sim\left(v_{1},v_{2}\right)=\cos\left(\theta \;\right)=\frac{v_{1}\cdot v_{2}}{∥v_{1}∥∥v_{2}∥}=\frac{\sum_{i=1}^{n}v_{1i}v_{2i}}{\sqrt{\sum_{i=1}^{n}v_{1i}^{2}}\sqrt{\sum_{i=1}^{n}v_{2i}^{2}}}$$
where $v_{1i}$ and $v_{2i}$ are components of vector $v_{1}$ and $v_{2}$, respectively. The resulting similarity ranges from −1, meaning exactly opposite, to 1, meaning exactly the same, with 0 indicating that they are uncorrelated, and in-between values indicating intermediate similarity or dissimilarity.

For the selection of semantic similar cavity, the previously defined cosine similarity be used to find out the top maximum similarity surrounding cavities in the cavity embedding space. Here, we have an exploration of semantic similar cavities also similar in attributes and geometry. The number of cavities is quite large. If all cavity be chosen to visualize and analyze, the work will be heavy and repeatable. Therefore, we choose some typical cavities to sample the data. First, the cavity with an id of 2 is chosen as an example. Then, we calculate the cosine similarity of all cavities to the cavity with an id of 2. To explore the relationship between the semantic similarity of the cavity and the cavity attributes and geometric structures, a different threshold must be set for cosine similarity of the cavity. One threshold is a similarity larger than 0.9, and the other threshold is a similarity between 0 and 0.1. Finally, when the threshold is larger than 0.9, 13 cavities are selected from 8052 cavities. When the threshold is greater than 0 and less than 0.1, 9 cavities are selected.

The length–width view be used for the purpose of exploring the relationship between cavity semantic information and cavity properties. As shown in Figure 6A, the length–width view takes the length as the X axis and the radius of the cavity as the Y axis. This representation can help users comprehend the entire cavity from the active site to the outer surface, particularly the positions of bottlenecks. Comparing Figure 6A with Figure 7A, there are many differences: The cavity with an id of 2, and its high similarity cavities in Figure 6A, has a similarity structure. The trajectories of the various cavities are roughly the same. This view clearly shows the consistency of cavities changes over time. However, the Figure 7A with low similarity has a chaotic view. At some points on the X axis, some of the cavities have the widest width and the others have the narrowest width.

To represent the cavity intuitively, the 3D geometric structure view be used to show the true location of cavities; there are clear cavity distribution characteristics, see Figure 6B and Figure 7B. Cavities be selected with the threshold is larger than 0.9 in Figure 6B geometric structure and spatial distribution are consistent. In Figure 7B, we can find cavities with the threshold is between 0 and 0.1 geometry are different: there are branches.

Through the comparative analysis mentioned above, we discover semantic similar cavities that are also similar in attribute and geometry. By comparing the different thresholds of the cavity with an id of 2, it can be concluded that the larger the threshold setting, the greater the similarity of various properties. In our paper, we only consider the cavity structure information (analogous to the semantics of the text) when training data: we do not consider its actual spatial position information. If the similarity threshold is set too small, there may be a situation that the actual spatial distance is far because the cavity semantic similarity is low, as Figure 7B shows.

To analyze clusters of semantically similar cavities, we need to cluster cavity trajectories. The hierarchical clustering is applied to cluster the 8052 cavity trajectories represented by the vectorization; 330 clusters were clustered. Through exploratory analysis of clustering results, some valuable information can be found, such as the main amino acid distribution at the molecular cavity bottleneck. By further analysis of the clustering results, we find that there is a similar topological properties in a certain kind of clustering results, but the real positions of the cavities are not the same. This is because our method vectorization representation cavity didn’t consider about the actual position of the cavity.

Clustering method can be used to analyze the pattern of a certain molecular cavity changes over time. To improve the accuracy of the clustering, we use the cavity data that have been classified according to the actual position as an input. The cluster1 (CAVER clustered results) be taken as an example which is the set of a certain cavity trajectories changes over time. The cluster1 contains 500 cavity trajectories. To judge how many patterns exist in cluster1, the t-SNE visual dimension reduction method was used to process it. From the t-SNE view Figure 8, it can be found that cavities should be clustered into three clusters. Then, the K-means method was used to classify cluster1 into three clusters.

To have a better analysis of the clustering results, we design some charts. For examples, Figure 9 is the length–width view of cluster1, cluster00, cluster01, and cluster02. From this view, the radius of the cavity varies with length of the active site. Figure 10 is the 3D geometric structure view of cluster00, cluster01, and cluster02. From this figure, the actual location and geometry of each cavity in the protein can be known. From Figure 9 and Figure 10, it can be seen that our clusters are more organized and show a good clustering result. The cluster1 mainly contains two major behavior patterns. One type includes cluster00 and cluster01, the other type is cluster02. Cluster02 is significantly different from cluster00 and cluster01. The bottleneck of cluster 1 is mainly reflected in cluster02. If we only need to analyze the bottleneck of the cavity, we must pay more attention to cluster02. Using our method can reduce the amount of work required to find the cause of the cavity bottleneck. If users need to pay attention to the differences between the different clusters, it is necessary to introduce the amino acid map.

To further analyze the cavity amino acid composition in each pattern, the type and number of amino acids contained in each cavity of cluster1 is designed to shown in bar view. As Figure 11 shows, the cavity amino acid distribution of cluster02 in HID is significantly different from the other two clusters. After screening, the amino acids that cause the cavities to differ can be identified, see Figure 12. The number of amino acid HID in cluster02 is half of cluster00 and cluster01. The number of TRP in cluster01 is half of cluster00 and cluster02. We can speculate that the differences in the pattern of cavity changes were caused by the different amino acids.

From this experiment, our method can be proved that it can maintain the topology characteristics of the cavity and can find the change patterns of a certain cavity.

## 4. Discussion and Conclusions

In this paper, a distributed representation method is proposed for protein cavity data. This method uses the word embedding technique to represent the cavity path, providing a new perspective for analyzing the sequence of cavities. Compared with the traditional data statistics based method, this method has two advantages: (i) Providing an abstract representation of molecular cavities, which facilitates users to perform operations, such as querying, comparing, and classifying, allowing discovery of some hidden features. Compared with traditional methods, this method can discover some hidden patterns. In addition, the word vector method is used to represent molecular cavities. The cavity is easier to use and other techniques such as machine learning. (ii) Using molecular cavity data to establish a connection between the cavity lining atoms; this connection contains the topological features of the cavity, which enables researchers to discover more cavity change patterns based on this connection, who otherwise may not be able to discover them.

To explore and verify the usefulness of our method, we use a real dataset to demonstrate the effectiveness of our method. We can know that semantic similar cavities also similar in attribute and geometry and molecular cavity can cluster by semantic similarity, can find the change patterns of different cavities.

The method is based on the molecular cavity lining atomic sequence. Emphasis was put on the connection between atoms and ignoring the properties of the cavity itself, such as the actual position and width of the cavity, as it can cause errors. In the future, our work will focus on combining other properties of the molecular cavity to enhance the presentation ability. We intend to further extend the work to design a more general visual analysis system for druggability analysis, which is an interesting direction for future work.

## Figures and Tables

**Figure 1 ijms-20-06019-f001:**
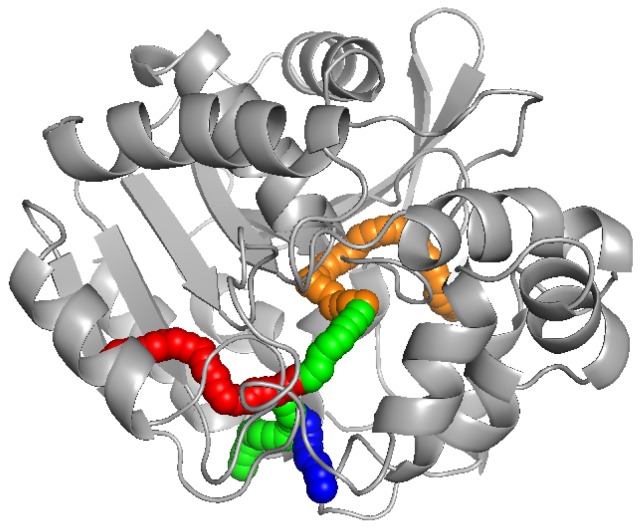
Set of cavities computed by CAVER and visualized using software PyMOL. The protein molecule is shown as a gray cartoon representation.

**Figure 2 ijms-20-06019-f002:**
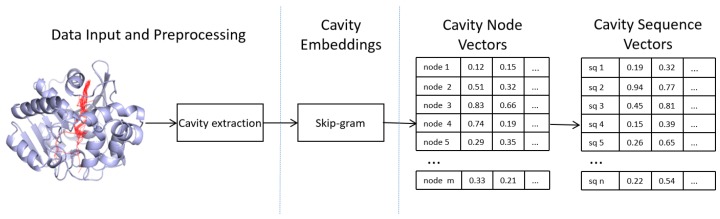
An illustration of generating molecule cavity distributed representations pipeline. Step 1: We calculate and extract molecule cavity sequences. Step 2: Cavity embeddings (i.e., vector representations of cavity nodes) are generated via a Skip-gram model. Step 3: Generate cavity sequence vectors from cavity node vectors.

**Figure 3 ijms-20-06019-f003:**
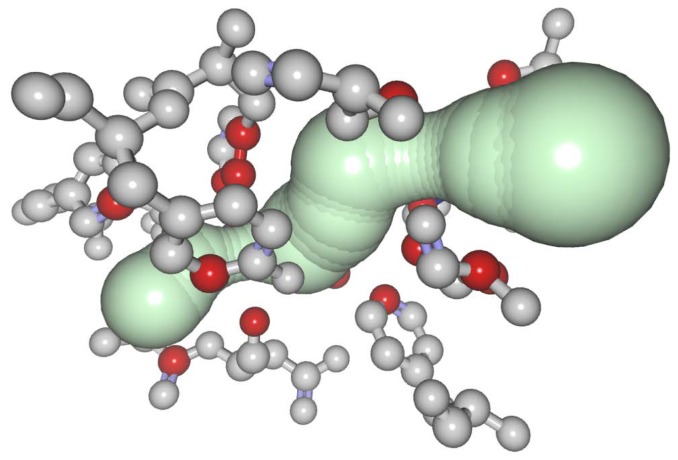
Cavity-lining residues (gray) with cavity-lining atoms (red) for the main cavity of the DhaA haloalkane dehalogenase (PDB ID 1CQW). In this case, the atoms and residues are visualized using Balls and Sticks method. Image taken from [29].

**Figure 4 ijms-20-06019-f004:**
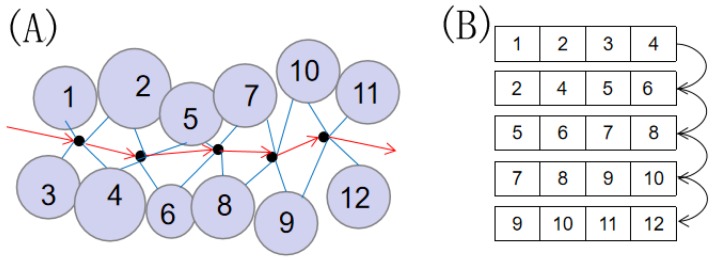
(**A**) The illustration of lining atoms. The closest four Surrounding atoms to each cavity sphere is lining atoms. (**B**) An example of a sequence of cavities.

**Figure 5 ijms-20-06019-f005:**
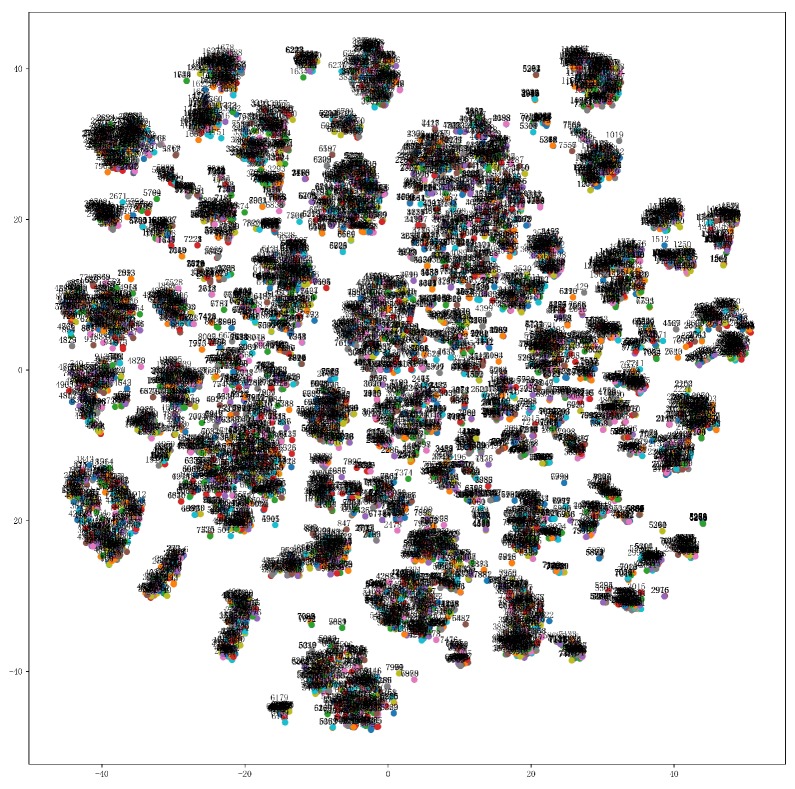
An overview of a cavity embedding space. t-SNE plot is obtained from 8052 cavity trajectories. It can be found that cavities should be clustered into hundreds of clusters. By comparison with the clustering result of CAVER Analyst 2.0 [28], which includes 246 clusters, it can be concluded that the vectorized representation of the cavity has the ability to represent the cavity.

**Figure 6 ijms-20-06019-f006:**
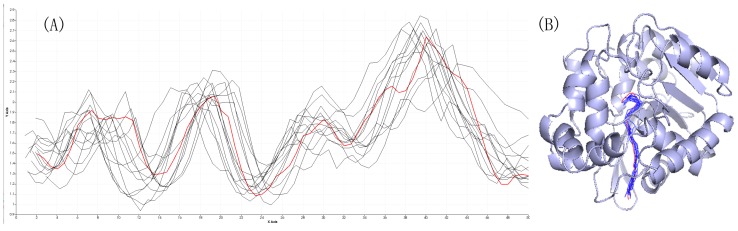
(**A**) The length–width view of the cavity with an id of 2 and its similarity cavities when the threshold is greater than 0.9. The red line is the cavity with an id of 2, these black lines are similar cavities with the cavity with an id of 2. (**B**) The 3D geometric structure view of the cavity with an id of 2 and its similarity cavities when the threshold is greater than 0.9.

**Figure 7 ijms-20-06019-f007:**
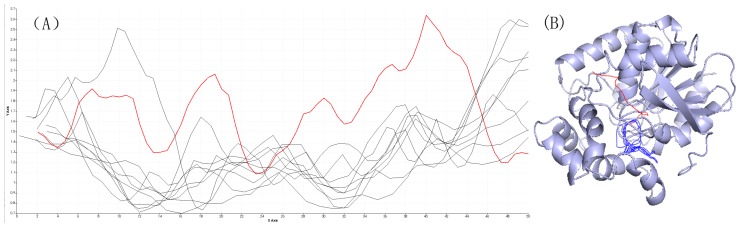
(**A**) The length–width view of the cavity with an id of 2 and its similarity cavities when the threshold is between 0 and 0.1. The red line is the cavity with an id of 2, these black lines are similar cavities with cavity id 2. (**B**) The 3D geometric structure view of the cavity with an id of 2 and its similarity cavities when the threshold is between 0 and 0.1.

**Figure 8 ijms-20-06019-f008:**
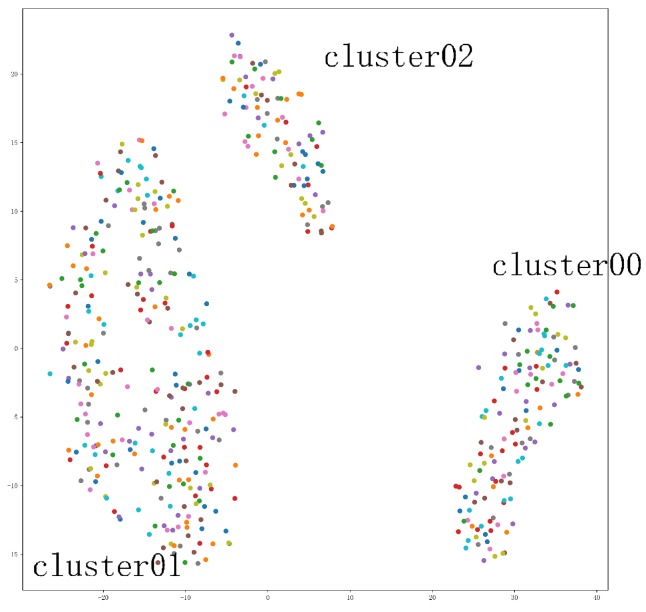
The t-SNE view of cluster1. t-SNE plot obtained from 500 cavity trajectories. It can be found that cavities should be clustered into three clusters.

**Figure 9 ijms-20-06019-f009:**
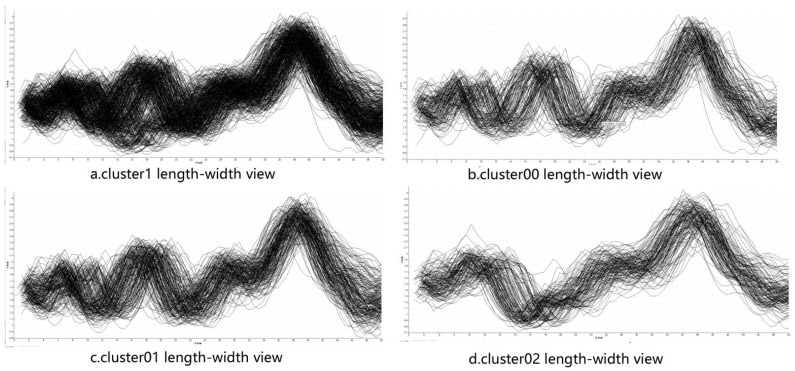
The clusters length–width view: (**a**) cluster1; (**b**) cluster00; (**c**) cluster01; (**d**) cluster02; this view was used to show the difference in cavity bottlenecks among different clusters. Cluster02 is significantly different from cluster00 and cluster01. The bottleneck of cluster 1 is mainly reflected in cluster02.

**Figure 10 ijms-20-06019-f010:**
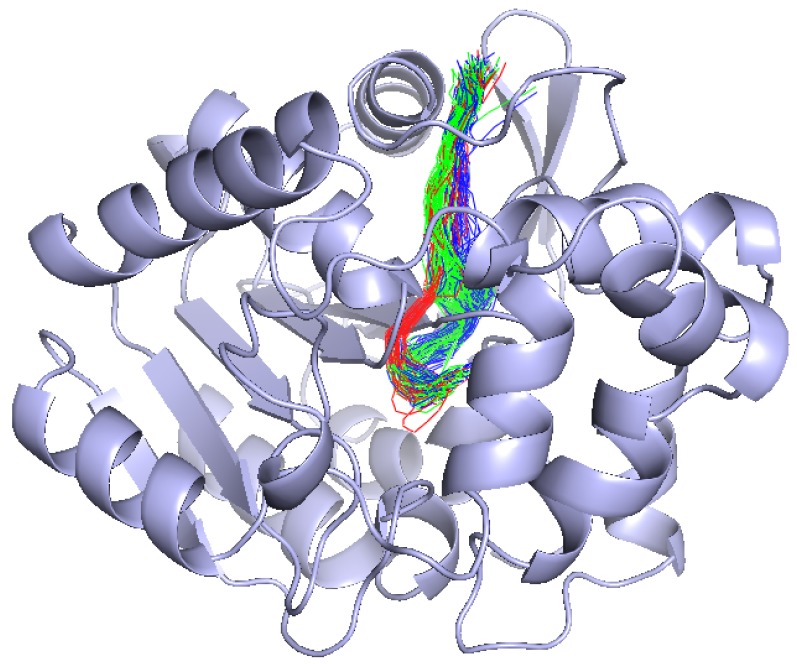
The 3D geometric structure view of cluster00, green; cluster01, blue; and cluster02, red. The actual location and geometry of each cavity in the protein can be shown.

**Figure 11 ijms-20-06019-f011:**
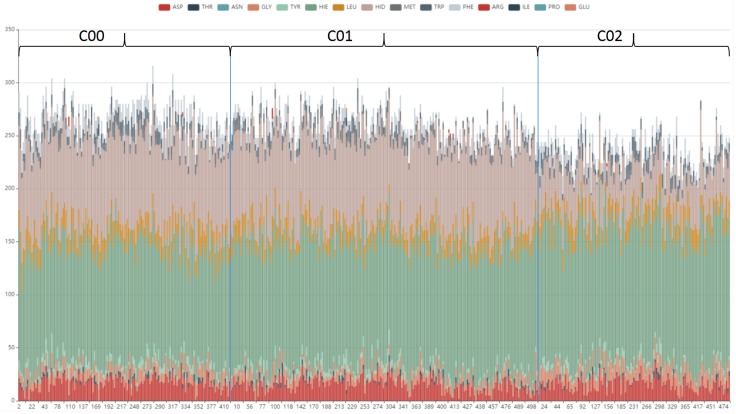
A bar view: type and number of amino acids contained in each cavity in cluster1, used to show the distribution of amino acids in different patterns. The cavity amino acid distribution of cluster02 in HID is significantly different from the other two clusters.

**Figure 12 ijms-20-06019-f012:**
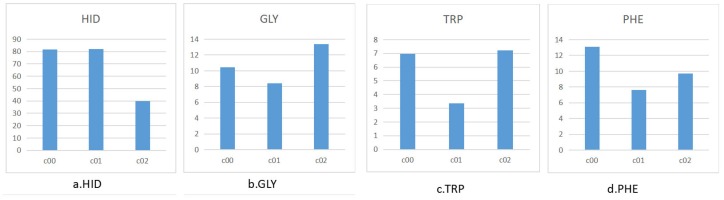
Average number of specific amino acids of each cavity in cluster00(c00), cluster01(c01), and cluster02(c02).

**Table 1 ijms-20-06019-t001:** Comparison of two main clustering methods.

Algorithms	Advantage	Disadvantage
K-means algorithm	simple and fast	the number of clusters should be known in advance
hierarchical clustering	the number of clusters does not require in advance	Computational complexity is too high

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
