# Peer review of "Molecular Cavity Topological Representation for Pattern Analysis: A NLP Analogy-Based Word2Vec Method"

_ijms, 2019, doi:10.3390/ijms20236019_

Round 1

Reviewer 1 Report

It would have been nice if authors clearly mentioned how they extract the sequence for a cavity is important in methods. I know they have explained different methods for cavity extraction but they have not mentioned specifically which they used for their study.

In the Cavity classification section, authors could have included small flow chart of the importance of two clustering methods. In this way readers can easily go through the difference between two main clustering methods.

Reviewer 2 Report

The authors propose an interesting new NLP-inspired approach to classifying dynamics voids in protein structures. The authors test this method out on cavities obtained from an MD trajectory of DhaA80. The idea and analyses in this manuscript are novel and interesting enough for me to recommend publication. I hope the authors address the following questions to improve the manuscript:

1. What do the authors mean by "dynamic pattern of molecular cavities"? Please do a better job of explaining the objectives of this work.

2. I am unable to glean any information from the t-SNE plot. Could the authors compare this visualization of cavity trajectories to other visualizations (such as PCA?)

3. Could the authors use their methods to provide insights into the druggability of the various classes of cavities they found?

There are a number of english/grammar issues:
"Natural Language Processing" is conventionally abbreviated as NLP, not NPL.
7: "demonstrate that 'our' approach"
12: "life function" ?
13: "macromolecular often requires *that* a small molecule ..." missing *that*
15: Remove examples. completely unnecessary.
20: "... can be *seen* in Figure 1."
27: "... individual *cavities* ...", "... *dynamics*, ..."

There are too many grammatical errors. This manuscript needs a lot of cleanup.

Reviewer 3 Report

The paper can be published in the actual format.

Only minor spell checking required.

Author Response

Point 1:  The paper can be published in the actual format. Only minor spell checking required.

Response 1: Thanks for your valuable suggestions. We have checked and polished the language of the full-text once again.